# Lived experiences and coping strategies of people living with Glaucoma in Nigeria: A qualitative study

Affiong Andem Ibanga[1], Nnaemeka Meribe[2], Bernadine Nsa Ekpenyong[3], Kelechukwu Enyinnaya Ahaiwe[4]*, Elizabeth Dennis Nkanga[1], Dennis George Nkanga[1], Uchechukwu Levi Osuagwu[5]

1 Department of Ophthalmology, Faculty of Clinical Sciences, University of Calabar, Cross River State, Nigeria, 2 Department of Politics, Media and Philosophy, La Trobe University, Melbourne, 3 Department of Public Health, University of Calabar, Calabar, Cross River State, Nigeria, 4 Department of Ophthalmology, University of Calabar Teaching Hospital, Calabar, Cross River State, Nigeria, 5 Bathurst Rural Clinical School, School of Medicine, Western Sydney University, Bathurst, New South Wales, Australia

* kcahaiwe@yahoo.com

## Abstract

### Background

The diagnosis of visual impairments and other chronic diseases is associated with frustration, anxiety and depression emanating from potential functional limitations, social restrictions, economic burden and future uncertainties. The impact of glaucoma extends across multiple domains of life. This study explores the lived experiences of people with glaucoma to understand the factors that influence their interactions within the community and how they cope with accompanying challenges.

### Method

This study utilised a qualitative research methodology. Data was collected using audio-recorded in-depth semi-structured interviews. Thematic analysis was used to analyse data collected through in-depth interviews.

### Result

Twelve participants aged 22–65 years old, mostly females (58.3%) participated in the study until saturation. Three major themes emerged from the data. The study identified psychosocial reactions to glaucoma diagnosis expressed with unpleasant emotions, fears and concerns about the future by persons living with glaucoma. Glaucoma impacted daily activities, routines and social life resulting in either reduced social activities or they completely stopped socialising. However, engaging in self-motivation and religious activities were useful strategies to cope with these challenges. Other adaptive and coping strategies included following eye care providers'

**Data availability statement:** All relevant data are within the manuscript.

**Funding:** The author(s) received no specific funding for this work.

**Competing interests:** The authors have declared that no competing interests exist.

**Abbreviations:** UCTH: University of Calabar Teaching Hospital; QOL: Quality of life; BD: Beck Depression scale; WHOQOL: World Health Organization Quality of Life questionnaire

instructions, attending clinic appointments, making reasonable adjustments at the home and work settings, transferring into accommodation that provides support needs as well as organizing itineraries and activities prior to leaving the house.

## Conclusion

This study provides evidence of the intricate psychosocial impacts of glaucoma, unravelling a spectrum of emotional reactions and lifestyle adaptations. The emotional stress, fears, and disruptions in daily life underscore the profound challenges posed by this chronic eye condition. Self-motivation and religious engagement emerged as potent coping mechanisms, highlighting the importance of psychological and spiritual support. Additionally, the study identifies practical strategies, from adhering to medical instructions to making environmental adjustments, offering valuable insights for healthcare and support networks.

## Introduction

Glaucoma is a group of conditions that can cause irreversible visual impairment and is characterised by progressive loss of retinal ganglion cells [1]. Elevated intraocular pressure (although not always elevated) is the only known modifiable risk factor [1]. Globally, glaucoma is the leading cause of irreversible blindness and the second leading cause of blindness [2]. In Nigeria, the prevalence of glaucoma is high (5%) among adults aged 40 and above. Furthermore, 94% of those with glaucoma are undiagnosed and untreated and one in five are blind [3,4]. Glaucoma, especially open-angle glaucoma is asymptomatic in the early stage, often leading to delayed diagnosis, late presentation and consequently delayed treatment [5]. Moreover, the biological basis is poorly understood and causes of the progression of the disease have not been fully characterised [6].

Diagnosis of chronic diseases such as glaucoma can be a cause of anxiety or depression, which is related to functional limitations, social isolation, loss of relationships, guilty feelings, anxiety about the future, heavy economic burden, and fear of potential blindness [7]. As the disease progresses, vision loss may set in and significantly impact the vision-related tasks' functioning and performance. Loss of vision is associated with some functional disabilities and may cause those with the disease to have difficulties in performing vision-related activities of daily living, including activities, like reading, driving, walking, climbing downstairs, various household chores, like sewing, cooking, fixing, etc. and limitations in social relations because of vision problems [8].

Response to visual loss by glaucoma patients varies widely. Various coping mechanisms have been documented, including making practical changes such as adjusting lighting, using handrails, and wearing magnifying glasses. Others actively made behavioural changes such as moving their head and eyes towards known areas of vision loss. Some patients sought support from friends and family while at the same time worried about becoming a burden. Still, others imposed self-restrictions or gave

up activities such as driving, thereby compromising well-being and independence [9]. Coping strategies such as seeking comfort in religious beliefs and practices have also been found to be beneficial to patients [10].

The use of some other coping methods, such as "substance use," may be termed "maladaptive" and may result in poorer health outcomes for the patients [11]. Other maladaptive coping strategies are self-blame and behavioural disengagement [11]. These are often underreported but need to be addressed by clinicians. Various factors might affect coping strategies and responses to visual loss. The type and quality of information received during clinical appointments, and the potential benefits of communication with other patients [9]. Studies have shown that culture and ethnicity can influence the use of coping strategies [12]. Several studies have documented the benefits of learning from the strategies, challenges, and successes of others (Davidson, 2016). Hope, a critical factor in healthcare [13], is also a key benefit, exposure to the narratives and experiences of others in similar situations [14].

Knowledge and awareness of coping and adaptive strategies will offer insight into the impact of the disease and patient care. However, not much is known about the personal stories of individuals living with glaucoma in Nigeria, as there is a paucity of qualitative studies investigating their lived experiences.

The present study aims to explore the lived experiences of people with glaucoma to understand the factors that influence their interactions within the community and how they cope with accompanying challenges. Findings from the study might also help inform future educational and management strategies for glaucoma. Indeed, findings from lived experience studies can be beneficial to other patients.

## Materials and methods

### Study design and setting

This study employed a qualitative research design. A qualitative design is generally considered the best method for exploring and describing people's views, particularly when the aim is to understand each person's unique perspective on any given issue [15]. The transcendental phenomenological approach was used to uncover the lived experiences of adults living with glaucoma in Nigeria. Transcendental phenomenology is a qualitative research approach that seeks to understand and describe human experience [16]. This approach focuses on the participants' descriptions rather than the researcher's descriptions. Thus, in this study, the researchers "bracketed" themselves to focus on gaining insight into the lived experiences of adults living with glaucoma in Nigeria [17].

This study consisted of semi-structured, in-depth, face-to-face interviews covering the experiences and coping strategies of the participants. All individuals who expressed interest and provided their contact information were reached by phone to confirm their eligibility. A total of eighteen potential participants expressed interest, and all were allowed to take part if eligible. In all, twelve (12) participants were recruited and provided with the participant information statement and consent form, and participated in the interviews till saturation was achieved. Saturation is a key concept in qualitative research that determines when data collection can cease based on the perceived adequacy or completeness of insight, while upholding the rigor and validity used to evaluate the adequacy of research data [18–20]. The study was conducted in Calabar, the capital city of Cross River State in southern Nigeria. The city is divided administratively into the Calabar Municipal Council and the Calabar South Local Government Area.

### Participants

Participants were recruited from the Out-Patients Department of Benita Eye Clinic, Zerah International Eye Hospital and Laser Centre, and the Eye Clinic of the University of Calabar Teaching Hospital, representing primary, secondary, and tertiary levels of eye care, respectively. Eligible participants were contacted to discuss the study's goals, procedures, and consent process before arranging an interview at a convenient time and location of the participant's choosing. Participants were screened to determine their eligibility. Eligible participants were aged 18 years and above, attended the clinic during the study period, diagnosed with and treated for glaucoma for not less than one year, and provided oral and written

consent to participate in the study. Exclusions included those under 18 years, glaucoma suspects, individuals with other ocular complications, or those experiencing conditions that hinder recall of any experience, such as Alzheimer's disease or dementia.

The study was described to participants as a one-on-one, in-person, semi-structured interview that would request information regarding their experiences as individuals living with glaucoma and their coping strategies. Ethical approval was obtained from the Human Research Ethics Committee of the University of Calabar Teaching Hospital (NHREC/07/10/2012). Data gathering was consistent with the awarding Ethics Committee Board and Helsinki Declaration requirements in studies involving human participants. Participants were assured of the confidentiality and anonymity of their responses.

## Data collection

Purposive sampling was used to recruit 12 participants (seven females and five males) from a possible pool of 100 potential participants. Purposive sampling is a non-probability sampling technique that, instead of aiming for statistical representativeness, focuses on identifying "information-rich" cases to provide in-depth insight into the phenomenon under study, following intentional participant recruitment based on specific characteristics or experiences relevant to the research question, particularly when investigating complex issues within public health contexts [21–25]. Participants were between 22 and 65 years old. Nine participants were university graduates, one was an undergraduate, and two were secondary school graduates. Interviews took place between 10 March 2022 and 3 June 2022. Before the interview, participants received the interview protocol and were invited to schedule a time to complete the interview. To ensure accessibility and increase participants' power in the interview process, the participants were offered to choose the location, read questions in advance, rearrange the protocol if they wished to, and skip or return to questions at any point during the interview. Participants were interviewed by one of the researchers (K.E.A.), who was trained and experienced in using the qualitative interview schedule. A pre-interview guide was used, and probes were employed during the interview to elicit further details and enhance meaning, as required. Interviews lasted for an average of 55 minutes and were audio-recorded using a digital recorder.

## Data analysis

Audio files were transcribed verbatim by one of the co-authors (K.E.A.), and transcripts were de-identified to maintain confidentiality (quotes from participants are referred to in this paper using codes). Only members of the research team had access to the transcripts. K.E.A. and another co-author (N.M.) read the transcript initially to have a fair understanding of each participant's personal story and become immersed in the data. Both researchers are experienced qualitative researchers. Thematic analysis was applied to the data using the six-step approach recommended by Braun and Clarke [26]. Line-by-line coding was adopted for each transcript. The two researchers independently reviewed the transcripts thoroughly and coded them to highlight the views of each participant. Emerging codes (words and phrases) were used to link similar statements between interviews. The researchers agreed that for a topic to become a theme at least two participants would have made substantive comments on the issue during their interviews. Thus, beyond just agreeing with each other, they also had to substantially discuss the topic based on their own experience [27].

## Catalytic validity and reflexivity

According to Williams and Morrow [28] standards of research quality cross paradigms, including "integrity of the data, balance between reflexivity and subjectivity, and clear communication of findings". Applying these components, the collection of rich data through in-depth interview transcripts and member checking helped the researchers to ensure the credibility of the present study. Furthermore, K.E.A., in alignment with the counsel of Jones, Torres [29], disclosed his position

and goals to the participants in conversations prior to commencing the interview with them and confirming their consent. K.E.A. shared his identity as a Nigerian working as an optometrist in the University of Calabar Teaching Hospital (UCTH), an individual not living with glaucoma and holding a master's degree in public health.

Regarding reflexivity, because of most of the co-authors' experiences as eye care professionals who have worked with many people living with glaucoma, the research team deemed it necessary to address concerns relating to bias and preconceptions concerning the phenomenon being studied. Thus, the researchers adopted the process of 'bracketing,' putting aside preconceptions and assumptions regarding the phenomenon being studied and remained open to participants' descriptions of their experiences of living with glaucoma [30].

## Results

### Demographic characteristics

The demographic characteristics of the participants are presented in Table 1. Participants were aged 22–65 years, all were Christians, the majority were females (58.3%), married (66.7%), and half of them lived in the urban centers. Nearly all participants had completed a university degree (83.3%), and a substantial proportion had glaucoma that was in the advanced stage (41.7%).

### Emerging themes

From the interview data, three themes emerged that described their experiences and coping strategies: (1) Psychosocial reaction to glaucoma diagnosis (2) Impact on everyday life (3) Coping with glaucoma.

Table 1. Demographic characteristics of the study participants.

| Code | Age (Years) | Gender | Marital Status | Highest Educational Level | Occupation | Status of City of Residence | Economic Status | Stage of Disease | State of Origin |
|---|---|---|---|---|---|---|---|---|---|
| G01 | 22 | F | Never Married | Undergraduate | Student | Semi urban | Fully Dependent | Advanced stage | Benue |
| G02 | 50 | F | Never Married | BSc Estate Surveyor | Civil servant | Urban | Partly dependent | End stage | Cross River |
| G03 | 46 | M | Married | LLB Law | Public servant | Urban | Independent | Advanced stage | Cross River |
| G04 | 43 | F | Married | SSCE | Business/ Petty Trading | Semi urban | Partly dependent | Moderate stage | Benue |
| G05 | 38 | F | Never Married/ Single Parent | BSc Edu Admin | Public servant | Semi urban | Independent | Early stage | Akwa-Ibom |
| G06 | 57 | F | Married | BSc Nursing | Public servant | Rural | Independent | Early stage | Edo |
| G07 | 50 | M | Married | MSc | Public servant | Urban | Independent | Moderate stage | Cross River |
| G08 | 65 | F | Married | BSc | Clergy | Urban | Independent | Advanced stage | Cross River |
| GO9 | 70 | M | Married | GCE | Subsistent/ Peasant Farming | Rural | Fully Dependent | End stage | Akwa-Ibom |
| G10 | 57 | F | Widow | BSc Nursing | Public servant | Urban | Independent | Advanced stage | Cross River |
| G11 | 52 | M | Married | MBBS | Lecturer/ Medical Consultant | Semi urban | independent | Moderate stage | Cross River |
| G12 | 65 | M | Married | LLB Law | Retired C/S | Urban | Partly dependent | Advanced stage | Cross River |

**Theme 1: Psychosocial reaction to glaucoma diagnosis.** Participants expressed sadness upon receiving their glaucoma diagnosis; for many, the lack of a cure was sufficient reason for dejection.

*I feel bad. I feel bad especially when I think of other people moving freely without any eye problem … When I heard that glaucoma had no cure, ah! When you have any sickness that has no cure, you will feel bad".*
*- (GP4).*

*"I felt bad. Honestly, I felt seriously bad. I was not happy at all. I felt bad. I was like, how can I become blind all of a sudden? Is that how I will go blind for the rest of my life at this age?* - (GP5)

Although some participants were aware that glaucoma diagnosis was not a death sentence and that vision may not deteriorate further with the appropriate management, yet they expressed unpleasant emotions following diagnosis:

*I went through a lot during that period. I even refused to do anything. I refused to talk because I was feeling down and was asking why I would have glaucoma? … So, it was very, very stressful that period. I was a little depressed. I refused to eat. I was always on my own, keeping to myself…I felt bad. I felt somehow because since my mother had it, my mind was that maybe one day I will also lose my sight.* -(GP10).

*Yes. Initially as a human being, I was depressed. I said, why me? I am one vibrant, always happy, young guy. So that shock… Depressed is the word I can only remember* (GP3).

One participant, GP8, said he contemplated committing suicide but was stopped by his son

*It wasn't easy…When my sight finally became bad, I sent my granddaughter to get me a bottle of bleach. I wanted to drink it. So, when she brought it, one of my sons asked her what she was doing with the bleach. And she said I was the one who sent her. He collected the bleach from her and took it away. I would have killed myself…* (GP2).

Some participants took the news with equanimity, regarding it as fate. For instance, GP2 said he was not surprised about the diagnosis but suspected that it was hereditary, as he had an uncle who was diagnosed with glaucoma. Nevertheless, GP2 considered it his fate as he was "*not worse than those who had gone blind. At least I can still drive, provided it is not at night.*" Similarly, some participants drew on their religious beliefs and awareness of medical sciences to accept the diagnosis as fate:

*After examining my eyes, the ophthalmologist mentioned that I had glaucoma. The word was scary, but I am a believer. She advised that I had better take care of my eyes or else I would go blind. For two days, I could not talk. I said God, how can that be?* (GP11).

Having the diagnosis of glaucoma established, most of the participants expressed fears and concerns about their future living with glaucoma. They worried about the possibility of losing their sight completely and consequently becoming a burden to family members.

*"I am worried because I don't know whether, as time goes by, I cannot do what I am doing because of reduced vision. So that…makes me feel afraid"* (GP4).

*"Haba! I am concerned about my future. I am deeply concerned about my future. I am concerned that if I don't take care of this sickness, it can lead to blindness and will affect my plans".* (GP5)

Some were more concerned about the high probability of their children and grandchildren also being diagnosed with glaucoma. For instance, GP7, a female nurse, said that she was worried about the hereditary nature of glaucoma:

*"I am concerned about my children and grandchildren. They should not be affected. It bothers me because it is hereditary…God should not allow them to be affected. Because if you become blind, it is not easy. Even now that I am not totally blind, it is not very easy with me because my field of vision is narrow; not spread as it is supposed to be. Before, I used to drive, but now, I cannot drive. I can't drive because of my vision. So, it gives me concern. It gives me concern".*

Similarly, another participant, GP6, noted:

*"But the only thing that disturbs me, though, is the fact that, you know, that glaucoma could be hereditary. So I can say that, a little bit, that I am worried about my children."*

Glaucoma diagnosis made some participants worry about their safety, as noted in this participant's response:

*"I worry about my safety. You could be in a place and there is danger around, but just because your vision is impaired due to glaucoma, you would not notice people running around or somebody with arms. And you will walk into danger. But if my eyes were sharper, because I would pick it from afar that that guy has a gun, I wouldn't need to walk close to him. I will run away"* - GP11

The ability to afford glaucoma medications was also a source of concern for some of the participants. While some of these participants can afford the medications now, they are worried that they may be unable to afford them in the future, as stated by a male civil servant:

*"At times, I ask myself if I would be able to afford my drugs when I retire. This is a question that confronts me every day. If I become retired, will I be able to buy my drugs? If it is so challenging to buy the medications now, I wonder what it would be like when I retire. That is the only thing. For one, I know I will not go blind, but how do I maintain my sight?"* - (GP12).

**Theme 2: Impact on everyday life.** According to the participants, glaucoma impacted their daily routines and social life. For instance, glaucoma made some participants feel useless and disorganised:

*"If not for my granddaughter now, it wouldn't be easy … I can go to the toilet by myself and do some other things… And after morning, my granddaughter goes to school, and until school closes and she is back, I cannot do anything for myself. So, it has affected me a lot. Everything stopped. I don't go out. I am no more useful to the family in any way. It is only advice that I provide, like I talk to them, maybe don't do this thing because of so, so and so. And it is only when I am told what happened that I can advise. But if I am not informed, I can't advise because I can't see what is happening around me"* - GP8.

Among the participants, there appeared to be a consensus that applying the eye drops at specific times of the day impacted their daily routines, often making it difficult to meet up with the regular chores:

*"It is quite discomforting. It is a problem of its own. It affects my schedule. At times, I will have to adjust my sleeping time to ensure I instill the eye drop. I will have to wake up again and ask my wife to help me out or I will do it myself. So*

*that inconvenience of having to wake up at a particular time to put the eye drop at night so that your eye pressure does not deteriorate is quite discomforting."*- (G03)

*"So that (instilling glaucoma drugs) interferes with my routine. If I go out and my eye drops are not with me by 8:00 pm, I rush back home. I try to get home before 8:00 pm, no matter how important what I have to do is. So sometimes it is like a sacrifice I make daily"*- (G07).

However, there were a few participants who said that their daily routines were not significantly impacted. For instance, GP6, related that:

*"I do not allow the issue that I have glaucoma to bother me. I wake up and do my routine. The only thing that changed is that I now adjust my time to be around to instill my eye drops when the time comes, and I will go out with the medication. You can say carrying the medication and stopping to instill the eye drop is the only way the glaucoma affects my routine."*

Similarly, GP10 expressed concern about the timing and frequency of instilling eyedrops, stating that:

*"What disturbs my daily routine the most is applying eye drops 3 times daily because I will apply it in the afternoon. For the eye drops applied twice daily, I don't have any problems with them. 6:00 am, I apply my drops, and 10:00 pm or 6:00 pm, I apply the eye drops again. So that one is not a problem for my routine because by then I will be at home. The one that I miss sometimes because of my daily activities is the eye drop I apply by 2:00 pm. I may be very busy, you know how the theatre is. I may be scrubbing a case inside the theatre when it is 2:00 pm".*

Indeed, glaucoma impacted the daily routines of all participants in one way or the other. While some considered the impact significant, others did not. However, those participants who did not consider it significant appeared to have worked out strategies that substantially mitigated the impact.

Another major aspect of life that living with glaucoma was the social life of the participants. Most participants highlighted that they could no longer participate in many activities they engaged in before they were diagnosed with glaucoma. Participants like GP2 said they were no longer able to drive under certain conditions, especially at night. For some participants, there was nothing like social life:

*"I don't work. I don't exercise. I don't go to social gatherings even in my compound. When people gather, you stay back. You don't go to anywhere; you will just be like a monument. I don't attend any other than church… my granddaughter takes me to church"*- GP9.

Remarkably, most of the participants who said glaucoma did not significantly impact their social life seemed to be those who never had much social life before they were diagnosed with glaucoma. For instance, G10 believed that glaucoma did not affect her social life as she still participated in church activities, but added that *"I don't participate in social activities."* This was also the case with GP11, who stated that:

*"Though I am not the outgoing type based on the discipline I got from my parents. I don't go out. Maybe I want to visit you, I will pass by your house. Or I have stayed for a long time, and I have not seen you. I will try to call, and the call fails, I will look for you. But I am not the type that goes out because I am staying with people who are undergoing training. That is my children. Social activities I attend is maybe old students' association meeting or church activities."* (GP11)

However, some participants like GP6, said they continued with their normal life since they had not been advised to avoid any social activity:

*"Since I don't know what social activities will worsen the condition, I just live my normal social life. The doctors or the nurses have not told me what will worsen the condition or what activities to disengage from. I don't know the things that will cause damage or worsen the glaucoma. So, I just live my normal life and take my medication."* (GP6)

Another participant, GP3, however, said that he reduced his attendance to social events because he felt embarrassed many times due to some of his friends who felt he was intentionally shunning them at events:

*"I am embarrassed at times by close friends who think that I intentionally snub them. They could be close to me, but because of my vision challenges, I would not see them. They usually think my action is intentional because my eyes are open and they are within an eyeshot of me, even waving at me. But I walked past. When we now meet at a closer range, they would wonder why I walk past them without exchanging greetings. I will then start apologizing and explaining my predicament".* (GP3)

From the foregoing, glaucoma impacted the social lives of most of the participants as they either reduced their social activities or stopped socialising. Only a few participants claimed that glaucoma did not affect their social lives. Interestingly, those who made such claims were mainly those who generally had little or no social lives before their glaucoma diagnosis.

**Theme 3: Coping with Glaucoma.** When asked to describe how they cope daily with the challenges associated with glaucoma, most of the participants said that following the instructions of eye care professionals and honouring clinic appointments have helped them to cope. GP11, for instance, understands that if he "could maintain my eye pressure by applying my eye drops daily, and my pressure is stable, then I will see the rest of my life. Failure of which I am not sure I will be guaranteed".

Likewise, other participants explained:

*"I also follow all the instructions of the doctor as given to me. Moreover, I am always here for my eye appointments. I don't miss it. That is all I can say".* (GP5)

*"I keep to the doctor's instructions of the drug application of three times daily. And I make sure I don't miss it"-* (GP8).

*"My strategy is to follow up with my appointment in the hospital and reserve money to buy my eye drops. Those are the only two things; follow up with my appointment and keep enough money to buy my eye drops."* - (G07).

Some participants further described that organizing their home and work settings in ways that would make life easier for them was very useful. This strategy for some involved transferring into different accommodations that meet his support needs:

*"Naturally, I am an organised person. So, with my lifestyle, I don't look for things in my room or anywhere. Because my things are well placed wherever I am. It is just like getting into your house in the dark. You know where this is, where this item is. I don't look for things. And anybody that enters the room, I tell them anything to see or touch, let them remain in the same position, please. So that I won't come and start looking for them later."* – (G02).

*"The method I use now is that things are kept at a particular place. Like this, my bag where my medication is, it is kept inside the wardrobe in one corner. So when I open the wardrobe, I just stretch my hand and pick up the bag for my medication. … now before dropping something, I will have to be very sure and know which part I kept it, and God gives me the memory to remember."* – (G09)

Similarly, some participants explained that they organize their itinerary and activities prior to leaving the house. They study the venue of events to determine the transportation and other support needs that may be required. For instance, GP3 explains:

*"I just make sure I set my itinerary at night to ensure I apply my eye drops at night. Then I make sure I plan the day's activity very well, church, work, etc. I try to get to functions very early and stay with people I am familiar with. I also avoid night activities completely".* (GP3)

Other participants gave a similar account, including GP2 and GP10, who stated that:

*"Before I go out, I put everything in place. And if I am calling a cab, I make sure I call someone one or two people around me will know. Yes. Maybe give them the mobile number and vehicle registration".* (GP2)

*"As I told you with regards to attending functions, I study where I go. If I don't know a place, I will not attend. That is the only strategy I can think of."* - (GP10)

Regarding financial matters, most participants disclosed that they manage their finances independently with minimal support. However, GP9, who has a significant reduction in vision, explained that he engages in limited financial transactions and relies on trusted relations to carry out financial transactions:

*"It is only my ATM that I am using. There is a cousin of mine who operates a POS. I take the card to him and I instruct him on the PIN to use and he does as I tell him."* (GP9)

Similarly, GP2 highlighted that she was always cautious in her financial dealings:

*"For a long while I didn't draw money myself. Yes, because it has to do with figures. I must see clearly before I transfer money to somebody".* (GP2)

Interestingly, one of the participants, GP1, a female civil servant, disclosed that not engaging in self-pity has helped her to cope. According to her:

*"Everything has to do with you yourself. If you want to wallow in self-pity, I pity you. Because in this our generation, you just must be up and doing. You encourage yourself. Like me as a person, there is nothing I hate like self-pity. You won't grow… Before I go out, I make sure everything I need is in place. And if I am calling a cab, I make sure I call someone - one or two people around me – so that they will know where I am going to. I can give them the mobile number and vehicle registration".* (GP1)

## Discussion

This study explored the lived experiences of people diagnosed and managed for glaucoma, providing insights into how glaucoma profoundly influences their lifestyle, community interactions, and coping strategies. A notable contribution of this research is that it focuses on a region largely unexplored in the context of glaucoma patient experiences, making it the pioneering study of its kind in this part of Nigeria. The findings resonate strongly with existing global literature on glaucoma patient experiences [31–33], emphasizing shared psychosocial responses among patients.

A key finding of this study is the significant psychosocial responses exhibited by the participants, who expressed distress and despondency upon receiving a glaucoma diagnosis. This aligns with established evidence that glaucoma patients commonly experience psychological stress when diagnosed with glaucoma [33]. This is manifested as heightened anxiety and depression, especially when confronted with the possibility of impaired vision, the lifelong treatment regimen, and the threat of irreversible blindness [32,34,35]. Notably, these findings mirror a phenomenological study conducted in China [33], indicating a global trend in the psychosocial impact of glaucoma diagnosis. Participants in both

studies shared a lack of awareness about the disease before diagnosis, resulting in heightened anxiety, frustration, powerlessness, a loss of independence, guilt, and fear of going blind, and concerns over the unpredictable progression of the disease and its genetic implications. Similarly, Odberg, Jakobsen [36] found that glaucoma diagnosis impairs the quality of life, attributable to the low knowledge of glaucoma among affected individuals.

In contrast to our findings, experiences varied among glaucoma patients in Turkey [31], where participants, diagnosed following acute and unrelated symptoms, expressed gratitude that the disease was not life-threatening. This divergence underscores the complexity of glaucoma experiences and the influence of cultural and contextual factors. In the present study, misconceptions persisted among participants who viewed glaucoma as incurable and expressed concerns that extended to suicidal ideation, highlighting the need for targeted education and support interventions.

Financial implications of glaucoma treatment and management emerged as a significant concern, with participants bearing the out-of-pocket costs of consultations and medications due to the inefficiency of the national health insurance scheme. Furthermore, they also bear the indirect cost, which may include transportation and travelling with an accompanying person. This resonates with broader discussions on the economic burden of glaucoma treatment, encompassing medications, diagnostic investigations, and surgery [37]. The long-term financial implications, particularly in the context of ageing and retirement, pose additional challenges to sustained glaucoma management, potentially leading to worsening disease conditions.

The economic burden associated with managing glaucoma in Nigeria poses a significant challenge, particularly when considered within the broader socio-economic context [38,39]. As of 2024, the minimum wage in Nigeria is approximately ₦30,000 (equivalent to USD 20) per month, which is barely sufficient to meet basic living expenses [40,41]. For many Nigerians, particularly those reliant on informal jobs or living in rural regions, this wage level remains difficult to achieve [42,43]. In their study, which compared the cost of medical versus surgical management of glaucoma in Nigeria, Omoti et al [44] found that the annual cost associated with medical management of glaucoma varied from ₦21,000, depending on the medication regimen and frequency of follow-up consultations. In comparison, surgical management requires a higher initial cost (approximately ₦30,000 per surgery), though considered more cost-effective more cost-effective in the long run due to minimized medication dependence. However, these figures must be examined within a regional context. For instance, tertiary eye care is more readily accessible in urban centres where specialists in glaucoma and diagnostic equipment are comparatively predominant. In contrast, rural and underserved areas, including specific settlements in northern Nigeria and the Niger Delta, may experience exacerbated costs due to travel expenses, accommodation, and time away from work, thereby rendering treatment even less affordable and accessible. This economic context underscores the lived experiences of many Nigerians living with glaucoma, who frequently encounter multiple challenges of managing the disease while coping with financial constraints. For individuals whose earnings are at or below the minimum wage, the expense associated with consistent glaucoma management may represent several months of income annually, resulting in poor adherence to treatment regimens and an increased risk of vision loss. These conditions highlight the urgent need for policy interventions to improve access to affordable eye care, subsidize glaucoma medications, and expand health insurance coverage to include chronic eye conditions.

The impact of glaucoma on daily living activities was profound, restricting participants' social engagement, physical activities, and even financial transactions due to reduced visual functions. According to Kalyani, Dayal (32), the progression of glaucoma is reflected in the deterioration of patients' visual functions, resulting in difficulty in executing activities of daily living, including driving, reading, and socializing. This also resonates with previous studies highlighting the pervasive influence of glaucoma on various aspects of life [45]. In this study, participants with glaucoma struggled considerably with engaging in activities of daily living. Engaging in social activities, especially during evenings and in unfamiliar environments, including attending wedding ceremonies, church activities, physical exercising, performing civic obligations, and driving, was challenging. Restrictions on social participation could be embarrassing and frustrating for individuals with glaucoma, resulting in reduced or abandonment of certain tasks and social lives [11,31,33]. Also, the treatment guidelines

for glaucoma consist of the administration of ocular medications at specific times of the day, and adherence to this routine can be disruptive to the individuals' daily plans, leading to frustration. As noted previously, the limitation of participation in activities or avoiding certain activities entirely, as found in this study, is a major source of worry [9,32]. In response to these challenges, participants employed adaptive strategies such as meticulous planning and environmental restructuring of home and work environment in ways that support their needs and make reasonable adjustments for efficiency to enhance their independence.

A noteworthy coping strategy was self-motivation, primarily derived from religious beliefs. Participants drew on their faith to navigate the challenges of glaucoma, showcasing the role of religion and spirituality as coping mechanisms. Religious belief inspired the faith and optimism that following management guidelines, including honouring clinic appointments, would help overcome the challenges of glaucoma. This finding agreed with existing literature, highlighting the role of faith in providing hope and facilitating adaptation to the challenges posed by glaucoma [33,46]. Importantly, the study did not identify maladaptive behaviours as coping mechanisms, differing from other research that reported venting, substance use, self-blame, and disengagement [11].

### Limitations and strengths of the study

The study was conducted in urban centres, and participants were patients with established diagnoses receiving treatment for glaucoma at these locations. Despite efforts to achieve a robust heterogeneity in the selection, the participants had attained at least a secondary education, which is above the basic education in Nigeria, and of a corresponding socioeconomic status, above the poverty line. These demographics may bias the results towards more positive coping strategies. Consequently, may not sufficiently reflect the experience of indigent, uneducated individuals in rural areas. The results may be different in this population. While participants in this study had been receiving glaucoma care for at least one year, we did not explore how experiences varied based on the duration since diagnosis. The duration since diagnosis could influence emotional adaptation, coping strategies, and health-seeking behaviour, and may provide further insight into the evolving nature of living with glaucoma. Future research could consider this variable to better understand how the trajectory of the condition impacts patients' psychosocial experiences over time. Despite the limitations, the study has numerous strengths, including its pioneering nature as a study, its diverse sample, and its qualitative design allowing for in-depth exploration of participants' experiences. Also, by identifying coping strategies, the study goes beyond clinical aspects and delves into the psychosocial and economic dimensions of glaucoma experiences. This holistic approach provides a more comprehensive view of the challenges faced by patients, contributing valuable information for designing targeted interventions and practical implications for intervention. By uncovering misconceptions, financial challenges, and adaptive strategies, this study provides practical insights that can inform interventions. These strengths collectively enhance the study's credibility, relevance, and potential impact on both academic and healthcare communities. However, further studies to explore the experience of uneducated indigent rural settlements may be rewarding.

### Conclusions

Glaucoma significantly impacts different aspects of life, including daily routines and social life. This study provided a comprehensive understanding of the lived experiences of glaucoma patients in Calabar, Nigeria, shedding light on psychosocial responses, financial challenges, and adaptive coping strategies. Patients received the diagnosis of glaucoma with anxiety, depression, and suicidal ideation primarily due to misconceptions about glaucoma. They were afraid of the blinding consequences, feeling unsafe, loss of income and independence, which they overcame by involvement in religious activities, adherence to treatment guidelines, attending clinic appointments, making reasonable adjustments at home and work settings, and organizing itineraries and activities in advance. The findings underscore the need for targeted interventions, including educational programs, improved access to affordable healthcare, and psychosocial support, to enhance the well-being of individuals managing glaucoma in this region.

## Recommendations

It is imperative to review the current glaucoma management protocol to incorporate tools for early psychosocial assessment and intervention. Strong emphasis should be laid on adequate patient education and counselling at the stage of glaucoma diagnosis, as well as raising awareness about the disease amongst the general population including family members of patients. Advocacy for the strengthening of health insurance and setting up of support groups should not be overemphasised. These efforts may contribute towards achieving a better quality of life for glaucoma patients.

## Author contributions

**Conceptualization:** Kelechukwu Enyinnaya Ahaiwe, Affiong Andem Ibanga, Nnaemeka Meribe, Bernadine Nsa Ekpenyong, Elizabeth Dennis Nkanga, Dennis George Nkanga, Uchechukwu Levi Osuagwu.

**Data curation:** Kelechukwu Enyinnaya Ahaiwe, Dennis George Nkanga, Uchechukwu Levi Osuagwu.

**Formal analysis:** Kelechukwu Enyinnaya Ahaiwe, Nnaemeka Meribe.

**Methodology:** Nnaemeka Meribe, Uchechukwu Levi Osuagwu.

**Supervision:** Dennis George Nkanga, Uchechukwu Levi Osuagwu.

**Validation:** Uchechukwu Levi Osuagwu.

**Writing – original draft:** Kelechukwu Enyinnaya Ahaiwe, Affiong Andem Ibanga, Nnaemeka Meribe, Bernadine Nsa Ekpenyong, Elizabeth Dennis Nkanga, Dennis George Nkanga, Uchechukwu Levi Osuagwu.

**Writing – review & editing:** Kelechukwu Enyinnaya Ahaiwe, Affiong Andem Ibanga, Nnaemeka Meribe, Bernadine Nsa Ekpenyong, Elizabeth Dennis Nkanga, Dennis George Nkanga, Uchechukwu Levi Osuagwu.

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
