## [Decision Letter · Decision Letter 0]

5 Aug 2024

PONE-D-24-11030Lived experiences and coping strategies of people living with Glaucoma in Nigeria -A qualitative study.PLOS ONE

Dear Dr. Ahaiwe,

Thank you for submitting your manuscript to PLOS ONE. After careful consideration, we feel that it has merit but does not fully meet PLOS ONE’s publication criteria as it currently stands. Therefore, we invite you to submit a revised version of the manuscript that addresses the points raised during the review process.

We look forward to receiving your revised manuscript.

Kind regards,

Aparna Rao 

Academic Editor

PLOS ONE

Reviewers' comments:

Reviewer's Responses to Questions

**Comments to the Author**

1. Is the manuscript technically sound, and do the data support the conclusions?

Reviewer #1: Yes

Reviewer #2: Partly

2. Has the statistical analysis been performed appropriately and rigorously? 

Reviewer #1: N/A

Reviewer #2: N/A

3. Have the authors made all data underlying the findings in their manuscript fully available?

Reviewer #1: Yes

Reviewer #2: Yes

4. Is the manuscript presented in an intelligible fashion and written in standard English?

Reviewer #1: Yes

Reviewer #2: Yes

5. Review Comments to the Author

Reviewer #1: Thank you for the opportunity to read the paper on the experiences and coping strategies of people living with glaucoma in Nigeria. The paper is important and brings forth the experiences of individuals with glaucoma. Here are a few suggestions to improve the paper:

1. Does the fact that the interviews were conducted with a group of patients from outpatient clinics in large hospitals affect the interviewees' group? Is there a population group that was not included in the interviews?

2. The first theme in the paper deals with psychological response. However, there are references to issues whose connection to psychological response is not clear, such as the ability of buying medications. Is the fear the psychological symptom? It was not sufficiently clear.

3. In the discussion, you raised the issue of the difficulties faced by glaucoma patients in several aspects and the lack of knowledge people have about the disease. Difficulty due to illness is natural, as is a lack of knowledge about the disease. However, and this is the main weakness of the paper, there is not enough discussion on possible ways to mitigate these difficulties. It is important to note how you believe the state or society in Nigeria can address the challenges raised in the study, whether from a social, economic, or medical perspective.

4. Regarding the economic issue - the economic difficulty was mentioned both in the results and in the discussion, but I did not fully understand the problem. How much does monthly glaucoma treatment cost in Nigeria? Is there public funding for this treatment? What are the amounts involved, and how does this impact the individual patient?

Overall, this is an important study, and I hope that after the revisions, the points I raised will become clearer.

Reviewer #2: Lived experiences and coping strategies of people living with Glaucoma in

Nigeria -A qualitative study.

Many thanks for the opportunity to review this interesting study. Using the COREQ Checklist I have identified a number of areas where this submission could be improved which I hope you will find helpful.

Introduction:

P1 Line 90 – fixing is highlighted as one of the activity limitations – can more clarification be provided here – fixing what? Some readers may be unfamiliar with this term in relation to domestic tasks.

P2 Line 115 missing word - critical factor in healthcare [13], is also a key benefit with exposure to narratives and experiences the

P 4 Line 117 – I understand that there has been little research in Nigeria on this topic, can you draw on research conducted in other areas for your background section?

Methods:

I would suggest that this is a qualitative study with the sample taken from a larger cross-sectional study (which is not apparent until the methods) as there is no inclusion of any references or methods relating to a phenomenological approach therefore I suggest removing reference to this study being a phenomenological study as highlighted in the abstract.

Line 132 conducted on adults – conducted with adults might be more appropriate terminology.

Line 135 A subsample of phase 1 participants were then invited to join the second phase of the study comprising of semi-structured, in-depth, face-to-face interviews covering the experiences and coping strategies of these participants – can you provide more information about how were this sample selected and recruited? Did you contact all 100 participants with glaucoma participating in the larger study? How were participants approached? Did any participants drop out?

Line 148 participants attended the clinic during the study period, - please provide the study period

Line 171 – participants could rearrange the protocol if they wished - do you mean change the order of the questions asked? Where was the location of the interviews conducted? Was anyone else present? Were any participants interviewed a second time or was all data captured with one interview each?

Line 174 - Pre-interview guide was used and probes employed during the interview to elicit further details and enhance meaning, as required – does this include the interview questions? Has this been provided as a supplementary file? Was this piloted prior to use? Did the researcher make field notes during the interview?

Line 189 - The researchers agreed that for a topic to become a theme at least two participants would have made substantive comments on the issue during their interviews. – does this method align with the Braun & Clarke approach to thematic analysis? What about the subsequent stages of the thematic analysis – how did the researchers refine and define themes? Was data saturation reached? Were transcripts or analysis returned to participants for checking? Did you use any software to assist with data analysis?

Line 193 – Thank you for your section on Catalytic validity and reflexivity – could you expand further to highlight if the research conducting the interviews already knew the participants?

Results:

Line 423 is repetition from line 389 – please revise your text.

Discussion:

Line 580 beginning with Hence doesn’t read very well can you please reword.

6. PLOS authors have the option to publish the peer review history of their article (what does this mean? ). If published, this will include your full peer review and any attached files.

**Do you want your identity to be public for this peer review?** For information about this choice, including consent withdrawal, please see our Privacy Policy .

Reviewer #1: **Yes: ** Rachel Nissanholtz Gannot

Reviewer #2: No

---

## [Author Response · Author response to Decision Letter 1]

29 Sep 2024

Response to reviewers’ comments

Dear Editor,

We appreciate the comments from the reviewers which has strengthened our study. Below, we have provided point by point responses to each of the comments. In the manuscript, the revised sections are highlighted in red fonts.

Reviewer #1

Thank you for the opportunity to read the paper on the experiences and coping strategies of people living with glaucoma in Nigeria. The paper is important and brings forth the experiences of individuals with glaucoma. Here are a few suggestions to improve the paper:

1. Does the fact that the interviews were conducted with a group of patients from outpatient clinics in large hospitals affect the interviewees' group?

Response: No. The recruitment of participants from this large hospital has no influence on interviewees group. Participants in qualitative research are not recruited to be representative of the population. This is because unlike quantitative studies, qualitative studies do not seek to generalize their findings (Levitt, 2021). The study explored the experiences of the participants concerning the subject under investigation.

Is there a population group that was not included in the interviews?

No. The targeted population was included. The study utilized a purposive sampling method to recruit its participants. According to Cambell et al. (2020), purposive sampling, a common technique in qualitative research is a non-probabilistic sampling method that involves deliberately selecting participants based on their relevance to the population characteristics and the study's objectives.

2. The first theme in the paper deals with psychological response. However, there are references to issues whose connection to psychological response is not clear, such as the ability of buying medications. Is the fear the psychological symptom? It was not sufficiently clear.

Response: Thanks for the comment. It appears that the reviewer may have misunderstood the study. The first theme deals with “psychosocial response” and not “psychological response.” Psychosocial responses are the emotional, cognitive, behavioural, and social responses that participants experienced in response to their glaucoma diagnosis. As seen in theme 1, there were emotional reactions of anxiety, fear and depression. Participants expressed concerns regarding the possibility of vision loss, leading to significant anxiety. For instance, the fear of spending their limited finance on medications also made them anxious. A participant also became depressed and wanted to commit suicide by drinking bleach. The fear of becoming a burden or the stigma associated with vision impairment also made some participants isolate themselves, withdrawing from social interactions. No change was made.

3. In the discussion, you raised the issue of the difficulties faced by glaucoma patients in several aspects and the lack of knowledge people have about the disease. Difficulty due to illness is natural, as is a lack of knowledge about the disease. However, and this is the main weakness of the paper, there is not enough discussion on possible ways to mitigate these difficulties. It is important to note how you believe the state or society in Nigeria can address the challenges raised in the study, whether from a social, economic, or medical perspective.

Response: Thanks for this comment. The discussion on possible ways to mitigate these difficulties have been included in the recommendation section of the manuscript. See lines 677-684 which reads like so:

The government at different levels in Nigeria can mitigate the challenges faced by people with glaucoma by improving access to affordable eye care services, especially in rural communities by establishing more eye care clinics and training additional ophthalmologists and optometrists. In addition, public awareness campaigns can be conducted to educate Nigerians about glaucoma, and the financial burden can be lessened by subsidizing the cost of medication and services, offering grants, or creating health insurance schemes. Furthermore, the government can liaise with local and international funding organizations to secure funding and resources for glaucoma research, treatment, and awareness programs.

4. Regarding the economic issue - the economic difficulty was mentioned both in the results and in the discussion, but I did not fully understand the problem. How much does monthly glaucoma treatment cost in Nigeria? Is there public funding for this treatment? What are the amounts involved, and how does this impact the individual patient?

Response: Thanks for this comment. We have addressed this in lines 595 to 605.

Glaucoma patients experience economic difficulty due to direct and indirect costs. The direct medical cost include cost associated with regular consultations for disease monitoring, h, investigations, treatment, and surgery. The indirect non-medical costs includes decreased productivity and employment opportunities, strained family finances, caregiver expenses, transportation, government purchase programs, guide dogs, and nursing home care (Varma et al. 2011). In Nigeria, majority of glaucoma patients are not subscribed to the National Health Insurance Scheme, and they fund their treatment out of pocket (Bello et al., 2023). The average monthly direct cost of glaucoma management in Nigeria was NGN 9,954 (about $6); and glaucoma patients can spend over one-tenth of their monthly income on glaucoma care alone (Bello et al., 2023).

Reviewer #2

Lived experiences and coping strategies of people living with Glaucoma in

Nigeria -A qualitative study. Many thanks for the opportunity to review this interesting study. Using the COREQ Checklist I have identified several areas where this submission could be improved which I hope you will find helpful.

Introduction:

P1 Line 90 – fixing is highlighted as one of the activity limitations – can more clarification be provided here – fixing what? Some readers may be unfamiliar with this term in relation to domestic tasks.

Response: Thanks for the comment. We have replaced the word “Fixing” with “repairs”

P2 Line 115 missing word - critical factor in healthcare [13], is also a key benefit with exposure to narratives and experiences the

Response: This has been revised and now read:

Hope, a crucial element in healthcare [13], is also a significant benefit gained from exposure to the narratives and experiences of others in similar circumstances [14].

*P 4 Line 117 – I understand that there has been little research in Nigeria on this topic, can you draw on research conducted in other areas for your background section?

Response: This has been discussed in the introduction. The relevant section in Lines 92 -110 now reads:

Few studies have examined the experiences of individuals living with glaucoma in Nigeria (Kyari et al., 2016; Abdull et al., 2016; Mbadugha & Onakoya, 2014). These studies have primarily focused on understanding the disease, access to care, delayed presentation for care, and the socioeconomic impact of living with glaucoma. Other studies have highlighted low glaucoma awareness, challenges in accessing care that lead to delayed presentation, and subsequent socioeconomic deprivation (Obasuyi et al., 2024; Okoye et al., 2018; Kyari et al., 2016; Onyia et al., 2022; Abdull et al., 2016; Mbadugha & Onakoya, 2014). The studies identified several factors contributing to limited access to glaucoma care, including individual-level factors such as inadequate education, unemployment, gender, rural residence, the cost of care, and religious beliefs (Obasuyi et al., 2024; Okoye et al., 2018). In addition, institutional-level factors, such as inadequate equipment and manpower shortages, significantly limit access to care, often leading patients to engage in risky help-seeking behaviours (Obasuyi et al., 2024; Okoye et al., 2018). Consequently, traditional healers have become the initial point of consultation for individuals experiencing reduced vision or those who do not notice improvement with treatment (Abdull et al., 2016).

Methods:

I would suggest that this is a qualitative study with the sample taken from a larger cross-sectional study (which is not apparent until the methods) as there is no inclusion of any references or methods relating to a phenomenological approach therefore I suggest removing reference to this study being a phenomenological study as highlighted in the abstract.

Response: Done. We have removed the reference to phenomenological study.

Line 132 conducted on adults – conducted with adults might be more appropriate terminology.

Response: Done. This has been revised to “conducted with”

Line 135 A subsample of phase 1 participants were then invited to join the second phase of the study comprising of semi-structured, in-depth, face-to-face interviews covering the experiences and coping strategies of these participants – can you provide more information about how were this sample selected and recruited? Did you contact all 100 participants with glaucoma participating in the larger study? How were participants approached? Did any participants drop out?

Response: Thanks for the comment. Although this was included in the article, we have made it clearer considering the questions you raised. The current section addressing these concerns now reads:

The study was conducted in two phases. In Phase 1, participants completed relevant questionnaires using validated measuring scales, and in Phase 2, the interviews were done. In the first phase of the study, survey respondents were given preliminary information about the second phase, which involved qualitative interviews. At the end of the survey, they were asked if they would be interested in participating in an interview and could indicate their interest by ticking a box in the questionnaire. Participants were then asked to provide their preferred contact details. All individuals who expressed interest and provided their contact information were reached by phone to confirm their eligibility. A total of eighteen potential participants expressed interest, and all were allowed to take part if eligible. In all, twelve (12) participants were recruited and provided with the participant information statement and consent form. A purposive sampling approach was employed to intentionally select participants based on their relevance to the population characteristics and the study's objectives.

Line 148 participants attended the clinic during the study period, - please provide the study period

Response: The study period has been provided.

……..attending the eye clinics Calabar, Nigeria, from 10 March 2022 and 3 June 2022.

Line 171 – Participants could rearrange the protocol if they wished - do you mean change the order of the questions asked? Where was the location of the interviews conducted? Was anyone else present? Were any participants interviewed a second time or was all data captured with one interview each?

Response: These details have been made clearer in the relevant section in the methods. The section now reads like this:

Participants were provided with the interview itinerary in advance but were given the option to choose the venue and time for the interviews. However, all participants preferred to have the interviews scheduled and conducted in the Resource Room of the Department of Ophthalmology at the University of Calabar Teaching Hospital, led by one of the researchers (KEA). The interviews were comprehensive, and all data was captured in a single session per participant, with no follow-up interviews conducted. Participants were interviewed by one of the researchers (KEA.), who was trained and experienced in using the qualitative interview schedule.

Line 174 - Pre-interview guide was used and probes employed during the interview to elicit further details and enhance meaning, as required – does this include the interview questions? Has this been provided as a supplementary file? Was this piloted prior to use? Did the researcher make field notes during the interview?

Response: These have been addressed.

The interview guide, which included the interview questions, was pre-tested with a cohort of individuals with high myopia attending the eye clinic to assess its suitability. However, this cohort was excluded from the final data collection. Probes were employed during the actual interviews to elicit further details and enhance the depth of responses as needed. Interviews lasted for an average time of 55 minutes and were audio-recorded using a digital recorder. The interview guide has been provided as a supplementary file. Additionally, the researcher took field notes during the interviews to capture contextual details and observations.

Line 189 - The researchers agreed that for a topic to become a theme at least two participants would have made substantive comments on the issue during their interviews. – does this method align with the Braun & Clarke approach to thematic analysis? What about the subsequent stages of the thematic analysis – how did the researchers refine and define themes? Was data saturation reached? Were transcripts or analysis returned to participants for checking? Did you use any software to assist with data analysis?

Response: The data analysis section has been revised to provide clarity on how the thematic analysis aligned with the Braun & Clarke approach, addressed the refinement of themes, explained the absence of member checking and software, and the researchers' perspective on data saturation.the relevant section inf the manuscript now reads like so:

Audio files were transcribed verbatim by one of the co-authors (K.E.A.), and transcripts were de-identified to maintain confidentiality (participant quotes are referred to in this paper using codes). Only members of the research team had access to the transcripts. Thematic analysis was conducted using the six-step approach outlined by Braun and Clarke [21]. To ensure data familiarization, K.E.A. transcribed the interviews verbatim. Both K.E.A. and co-author (N.M.), who are experienced qualitative researchers, thoroughly read the transcripts multiple times, line by line, to gain a deep understanding of each participant’s personal story and immerse themselves in the data.

During the initial coding process, the researchers agreed that for a topic to be identified as a theme, at least two participants must have made substantive comments on the issue during their interviews. Following this, the initial themes were refined and reviewed through an iterative process to enhance their clarity, coherence, and accuracy in conveying the overall meaning [21]. Finally, the themes were defined and named using descriptive labels that captured the core ideas they represented. Member checking was not performed, as the researchers felt that transcribing the interviews verbatim ensured an accurate representation of the participants' views. Given the diverse range of participant experiences, data saturation was not a primary consideration for this study. Data analysis was conducted manually without the use of software.

Line 193 – Thank you for your section on Catalytic validity and reflexivity – could you expand further to highlight if the researcher conducting the interviews already knew the participants?

Response: This has been clarified. The relevant section now reads:

The researchers were aware that prior familiarity with participants might undermine the credibility of research findings (Johnson et al., 2020). As a result, the researcher who conducted the interview was not familiar with any of the participants.

Results:

Line 423 is repetition from line 389 – please revise your text.

Response: Lines 422-426 are the summary of Theme 2: Impact on everyday life

Discussion:

Line 580 beginning with Hence doesn’t read very well can you please reword.

Response: “Hence” has been removed and replaced with: “As a result the findings may….”

---

## [Decision Letter · Decision Letter 1]

9 Dec 2024

PONE-D-24-11030R1

Lived experiences and coping strategies of people living with Glaucoma in Nigeria -A qualitative study.

PLOS ONE

Dear Dr. Ahaiwe,

Thank you for submitting your manuscript to PLOS ONE. After careful consideration, we have decided that your manuscript does not meet our criteria for publication and must therefore be rejected.

Specifically:

**ACADEMIC EDITOR: **

The sample size is seriously small, significantly heterogenous, not allowing any reliable conclusions to be made.The questionnaire followed was not a valid tested questionnaire cited in any referenceThe reporting of the results uses abstract and ambiguous terms with no quantitative information.While consideration of the patient's own words is priority in clinical examination, there is no place for such information in research with pooled information from a study sample or cohort.

I am sorry that we cannot be more positive on this occasion, but hope that you appreciate the reasons for this decision.

Kind regards,

Nader Hussien Lotfy Bayoumi, M.D., FRCS (Glasgow)

Academic Editor

PLOS ONE

Reviewers' comments:

Reviewer's Responses to Questions

**Comments to the Author**

1. If the authors have adequately addressed your comments raised in a previous round of review and you feel that this manuscript is now acceptable for publication, you may indicate that here to bypass the “Comments to the Author” section, enter your conflict of interest statement in the “Confidential to Editor” section, and submit your "Accept" recommendation.

Reviewer #2: All comments have been addressed

Reviewer #3: (No Response)

2. Is the manuscript technically sound, and do the data support the conclusions?

Reviewer #2: (No Response)

Reviewer #3: No

3. Has the statistical analysis been performed appropriately and rigorously? 

Reviewer #2: (No Response)

Reviewer #3: No

4. Have the authors made all data underlying the findings in their manuscript fully available?

Reviewer #2: (No Response)

Reviewer #3: No

5. Is the manuscript presented in an intelligible fashion and written in standard English?

Reviewer #2: (No Response)

Reviewer #3: Yes

6. Review Comments to the Author

Reviewer #2: Many thanks for addressing the suggested comments. I can see that changes have been made throughout the submission but the abstract still refers to a phenomenological study - can I suggest that this is changed also?

Reviewer #3: • L110: “The type and quality …..” is a sentence without verb.

• L114: The reference is alphabetic not numerical as the rest of the manuscript.

• L150: The criteria of diagnosis of glaucoma and its stage need to be mentioned in the inclusion criteria.

• L151: those under 18 years and glaucoma suspects cannot be exclusion criteria as they were not available according to inclusion criteria.

• L163: Sample size needs to be justified according to the primary outcome by proper sample size calculation referring to the source of the variance used in the formula.

• L166: The criteria of participants should appear in the results not the methodology.

• The questionnaire used in the study needs to be mentioned with support of its validation.

• The outcome of the study needs to be clearly defined in the methodology.

• The statements of the patients do not add to the value of the manuscript. They need to be represented in a measurable pattern and statistically analyzed to reach a valid conclusion.

• L422-426: The terms “most” and “few” are not appropriate for results presentation.

• The discussion and conclusion are a review of literature and opinions of the authors. They are not supported by statistically significant data in the results.

7. PLOS authors have the option to publish the peer review history of their article (what does this mean? ). If published, this will include your full peer review and any attached files.

**Do you want your identity to be public for this peer review?** For information about this choice, including consent withdrawal, please see our Privacy Policy .

Reviewer #2: **Yes: ** Katie Thomson

Reviewer #3: No

- - - - -

---

## [Author Response · Author response to Decision Letter 2]

24 Feb 2025

Dear Editor,

Thank you for your comments on our manuscript, “Lived experiences and coping strategies of people living with Glaucoma in Nigeria – A qualitative study” (PONE-D-24-11030R1). We deeply appreciate the time and effort you and the reviewers dedicated to evaluating our submission.

Upon reviewing the feedback, it seems that the comments address concerns more relevant to a quantitative study than the qualitative design and methodology employed in our manuscript. We believe this may have influenced your decision regarding our submission.

We would like to clarify key aspects of our study:

Sample Size: Our sample size aligns with best practices for qualitative research, as studies employing semi-structured interviews typically reach data saturation within 9–17 interviews (Hennink & Kaiser, 2022).

Data Collection Method: Our methodology relied on semi-structured interviews to capture the lived experiences of participants, not questionnaires, which are more common in quantitative studies.

Focus and Methodology: The aim of our study was to explore participants' unique lived experiences and coping strategies, which are best examined through qualitative analysis rather than through quantitative measurement. Consequently, the absence of numerical data is inherent to our methodological approach.

Patient Narratives: The patient narratives at the heart of our study are central to understanding the phenomena explored. This approach ensures rich, in-depth insights that quantitative methods would not capture.

Given these clarifications, we ask you to reconsider our manuscript to ensure the feedback aligns with its qualitative nature and research aims. If any additional information or clarifications are needed, we would be more than happy to provide them.

We sincerely thank you for your consideration and look forward to your response.

Kind regards,

Dr Kelechukwu Ahaiwe

---

## [Editor Report · Decision Letter 2]

10 Mar 2025

PONE-D-24-11030R2Lived experiences and coping strategies of people living with Glaucoma in Nigeria -A qualitative study.PLOS ONE

Dear Dr. Ahaiwe,

Thank you for submitting your manuscript to PLOS ONE. After careful consideration, we feel that it has merit but does not fully meet PLOS ONE’s publication criteria as it currently stands. Therefore, we invite you to submit a revised version of the manuscript that addresses the points raised during the review process.

**Please find below, comments regarding your responses and action.**

We look forward to receiving your revised manuscript.

Kind regards,

Osamudiamen Cyril Obasuyi, MD, MSc, FWACS, FMCOPh

Academic Editor

PLOS ONE

Journal Requirements:

Additional Editor Comments (if provided):

Materials and Methods:

1. This was a cross-sectional study conducted with adults living with vision impairment due to glaucoma, high myopia, diabetic retinopathy and low vision

a. It is important that the methodology and research paradigm of this work is stated clearly. As is, the opening lines gives the impression that this is a quantitative approach to the RQ proposed by the authors.

b. It may seem that this is part of a larger work, however, this paper focussed majorly on Glaucoma. The authors may need to focus their work on Glaucoma.

2. In Phase 1, participants completed relevant questionnaires using validated measuring scales

a. What were these scales? Please state these scales and include references.

b. How did these scales answer the research objectives of this paper?

c. Were these scales validated in the population being studied before use?

3. A total of eighteen potential participants expressed interest, and all were allowed to take part if eligible. In all, twelve (12) participants were recruited and provided with the participant information statement and consent form.

a. Could the authors state why there was a 30% attrition rate in response to phase 2?

4. Purposive sampling facilitates better matching of the sample to the aims and objectives of this research, enhancing the integrity of research

a. How was purposive sampling instrumental to achieving the aims of the study when

i. There were more females than males

ii. The participants were Christians

iii. The participants were well educated

b. Even though this was highlighted as a limitation, was there a better approach? Could the authors have been more specific in their recruiting process, seeing as qualitative studies allow for such sampling methods?

RESULTS

1. Most participants disclosed that they felt sad and despondent on receiving the news of their glaucoma diagnosis.

a. At what stage did these patients receive their diagnosis? Was this explored?

i. The authors stated that they discussed with patients who had advanced glaucoma

ii. The inclusion criteria was that the participants would have been receiving Glaucoma care for a year.

The above points presuppose that some participants may have been diagnosed longer than a year. If this is the case, exploring how they coped with the evolution of their disease provides more context to the discussion of the themes.

DISCUSSION

1. In Nigeria, the majority of glaucoma patients are not covered by the National Health Insurance Scheme and must pay for their treatment out of pocket [33]. The average monthly direct cost of managing glaucoma in Nigeria is NGN 9,954 (approximately $6), meaning patients may spend over one-tenth of their monthly income on glaucoma care alone.

a. The cost of glaucoma in Nigeria needs to be contextualised to provide clarity and understanding viz:

i. Minimum wage

ii. Regionally. See Omoti AE, Edema OT, Akpe BA, Musa P. Cost Analysis of Medical versus Surgical Management of Glaucoma in Nigeria. J Ophthalmic Vis Res. 2010 Oct;5(4):232-9.
---

## [Author Response · Author response to Decision Letter 3]

5 May 2025

Reviewer’s Comment Author’s Response Author’s Action

1

Materials and Methods:

1. This was a cross-sectional study conducted with adults living with vision impairment due to glaucoma, high myopia, diabetic retinopathy and low vision

a. It is important that the methodology and research paradigm of this work is stated clearly. As is, the opening lines gives the impression that this is a quantitative approach to the RQ proposed by the authors.

b. It may seem that this is part of a larger work, however, this paper focussed majorly on Glaucoma. The authors may need to focus their work on Glaucoma.

Thank you for the feedback and suggestions provided in your review. This study is part of a larger project that utilized a mixed-methods approach to explore various aspects of visual impairment, including diabetic retinopathy, high myopia, and low vision. However, this paper focuses exclusively on participants diagnosed with glaucoma and is entirely a qualitative study, employing a phenomenological approach. As such, the analysis, results, and discussion focus specifically on the themes derived from the experiences of participants living with glaucoma. The manuscript has been updated to reflect this focus in the introduction and methods sections. The manuscript has been updated to reflect this focus in the introduction and methods sections.

2. In Phase 1, participants completed relevant questionnaires using validated measuring scales

a. What were these scales? Please state these scales and include references.

b. How did these scales answer the research objectives of this paper?

c. Were these scales validated in the population being studied before use? In the first phase of the study, participants completed validated questionnaires to capture baseline information on their vision-related quality of life and psychosocial well-being. The following scales were used:

1. World Health Organization Quality of Life Scale (WHOQOL-BREF)

2. The Beck Depression Inventory (BDI) scale

However, these validated measuring scales were utilized for the quantitative study aspect of the larger study and provided contextual understanding of the participants' functional limitations and emotional states, which enhanced the qualitative study in Phase 2. It is not utilized as the primary focus of this manuscript's data analysis. However, the questionnaires guided the researchers in understanding participants' experiences and informing the selection of the semi-structured interview questions during the detailed interviews. Given this, the scale did not contribute to the study's qualitative part, nor the discussion and results sections of the manuscript. Therefore, the description of the validated scale has been expunged from the manuscript ‘Validated scales’ have been expunged from the manuscript since they were not used to collect qualitative data for this study.

3. A total of eighteen potential participants expressed interest, and all were allowed to take part if eligible. In all, twelve (12) participants were recruited and provided with the participant information statement and consent form.

a. Could the authors state why there was a 30% attrition rate in response to phase 2? Like in every qualitative research, our aim in this study was not to generalize findings to a larger population. Instead, we sought to understand experiences, perceptions, and social processes in-depth. Unlike quantitative studies, qualitative studies do not adhere to a predetermined number of participants. Instead, qualitative studies focus on achieving data saturation—the point at which no new themes or insights are being revealed from the data. In this study, saturation was achieved with the 12 participants who took part in the interviews. The definition of saturation has been included in the manuscript to improve clarity.

This was not an attrition rate, actually, but the proportion that consented for interviewed after responding to the quantitative arm of the study

Lines 51, 140-148

The saturation definition below has been included in the manuscript and further explanation given to improve clarity.

3. Purposive sampling facilitates better matching of the sample to the aims and objectives of this research, enhancing the integrity of research

a. How was purposive sampling instrumental to achieving the aims of the study when

i. There were more females than males

ii. The participants were Christians

iii. The participants were well educated

b. Even though this was highlighted as a limitation, was there a better approach? Could the authors have been more specific in their recruiting process, seeing as qualitative studies allow for such sampling methods? Thanks for this observation. This study employed purposive sampling to recruit participants diagnosed with glaucoma who possessed at least one year of treatment experience. This approach ensured that participants provided rich information and deep reflective insights that aligned with the study's objectives.

While this method facilitated alignment with the research goals, it was skewed towards females, Christians, and individuals with tertiary education, and these characteristics have been acknowledged as limitations within the manuscript.

Nevertheless, considering the study's exploratory nature and the context of urban hospital-based recruitment, the sample accurately represented the demographic engaging with glaucoma services in the study setting. A more targeted recruitment strategy—incorporating outreach through community eye clinics, religious organizations, and support groups for individuals with low literacy—might have resulted in a more representative population. Therefore, future research could focus on diversity and benefit from utilizing maximum variation sampling to ensure a broader representation of educational, religious, and gender backgrounds. Moreover, a definition of the concept has been included to the manuscript to further explain the selection process. A definition of the concept has been included to the manuscript to further explain the selection process:

recruiting the sample for focus groups and in-depth interviews. Qualitative health research, 11(1), 117-126.

2 RESULTS:

1. Most participants disclosed that they felt sad and despondent on receiving the news of their glaucoma diagnosis.

a. At what stage did these patients receive their diagnosis? Was this explored?

i. The authors stated that they discussed with patients who had advanced glaucoma

ii. The inclusion criteria was that the participants would have been receiving Glaucoma care for a year.

The above points presuppose that some participants may have been diagnosed for longer than a year. If this is the case, exploring how they coped with the evolution of their disease provides more context to the discussion of the themes. Thank you for this thoughtful comment. As stated, our inclusion criteria required participants to have been receiving glaucoma care for at least one year. However, we did not stratify participants based on the specific number of years since diagnosis. We acknowledge that the length of time since diagnosis can influence a patient’s lived experience and coping mechanisms, and this is an important area for further exploration. Accordingly, we have now included this point as a limitation in the revised manuscript and suggested it as a direction for future studies.

It is also worth noting that although participants may have varied in the duration since diagnosis, this variable did not emerge as a distinct theme in the data. Most participants consistently reported experiencing sadness and despondency upon initial diagnosis, regardless of the time since they were diagnosed. This suggests a shared emotional response to the diagnosis itself, which was more prominent in their narratives than the evolving nature of their condition over time.

Limitation revised to include this statement for future studies [lines 621-627]

3 DISCUSSION

1. In Nigeria, the majority of glaucoma patients are not covered by the National Health Insurance Scheme and must pay for their treatment out of pocket [33]. The average monthly direct cost of managing glaucoma in Nigeria is NGN 9,954 (approximately $6), meaning patients may spend over one-tenth of their monthly income on glaucoma care alone.

a. The cost of glaucoma in Nigeria needs to be contextualised to provide clarity and understanding viz:

i. Minimum wage

ii. Regionally. See Omoti AE, Edema OT, Akpe BA, Musa P. Cost Analysis of Medical versus Surgical Management of Glaucoma in Nigeria. J Ophthalmic Vis Res. 2010 Oct;5(4):232-9. Thanks for this suggestion. As requested, glaucoma management has been contextualised to reflect cost and regional implications in the discussion. The suggested reference was also added Lines 560-584

---

## [Editor Report · Decision Letter 3]

12 May 2025

Lived experiences and coping strategies of people living with Glaucoma in Nigeria -A qualitative study.

PONE-D-24-11030R3

Dear Dr. Ahaiwe,

We’re pleased to inform you that your manuscript has been judged scientifically suitable for publication and will be formally accepted for publication once it meets all outstanding technical requirements.

Kind regards,

Osamudiamen Cyril Obasuyi, MD, MSc, FWACS, FMCOPh

Academic Editor

PLOS ONE
---

## [Editor Report · Acceptance letter]

PONE-D-24-11030R3

PLOS ONE

Dear Dr. Ahaiwe,

I'm pleased to inform you that your manuscript has been deemed suitable for publication in PLOS ONE. Congratulations! Your manuscript is now being handed over to our production team.

Kind regards,

on behalf of

Dr. Osamudiamen Cyril Obasuyi

Academic Editor

PLOS ONE